# Oligometastatic Gastric Cancer: Clinical Data from the Meta-Gastro Prospective Register of the Italian Research Group on Gastric Cancer

**DOI:** 10.3390/cancers16010170

**Published:** 2023-12-29

**Authors:** Maria Bencivenga, Silvia Ministrini, Paolo Morgagni, Gianni Mura, Daniele Marrelli, Carlo Milandri, Maria Antonietta Mazzei, Mattia Berselli, Manlio Monti, Luigina Graziosi, Rossella Reddavid, Fausto Rosa, Leonardo Solaini, Annibale Donini, Uberto Fumagalli Romario, Franco Roviello, Giovanni de Manzoni, Guido Alberto Massimo Tiberio

**Affiliations:** 1General and Upper GI Surgery Division, Department of Surgery, University of Verona, 37125 Verona, Italy; maria.bencivenga@univr.it (M.B.); giovanni.demanzoni@univr.it (G.d.M.); 2Surgical Unit, Department of Clinical and Experimental Sciences, University of Brescia, ASST Spedali Civili di Brescia, 25100 Brescia, Italy; silvia.ministrini@hotmail.it; 3General and Oncologic Surgery, “Morgagni-Pierantoni” Hospital, 47121 Forlì, Italy; morgagni2002@libero.it (P.M.); leonardo.solaini2@unibo.it (L.S.); 4Department of Surgery, San Donato Hospital, 52100 Arezzo, Italy; gianni.mura@uslsudest.toscana.it; 5Unit of General Surgery and Surgical Oncology, Department of Medicine, Surgery and Neurosciences, University of Siena, 53100 Siena, Italy; daniele.marrelli@unisi.it (D.M.); franco.roviello@unisi.it (F.R.); 6Department of Oncology, San Donato Hospital, 52100 Arezzo, Italy; c.milandri@gmail.com; 7Unit of Diagnostic Imaging, Department of Medical, Surgical and Neuro Sciences and of Radiological Sciences, Azienda Ospedaliero-Universitaria Senese, University of Siena, 53100 Siena, Italy; mamazzei@gmail.com; 8General Surgical Unit I, Department of Surgery, ASST Settelaghi-Varese, 21100 Varese, Italy; mattia.berselli@asst-settelaghi.it; 9Department of Medical Oncology, IRCCS Istituto Romagnolo per lo Studio dei Tumori (IRST) “Dino Amadori”, 47014 Meldola, Italy; manlio.monti@irst.emr.it; 10Chirurgia Generale e d’Urgenza, Azienda Ospedaliera di Perugia, 06121 Perugia, Italy; luiginagraziosi@yahoo.it (L.G.); annibale.donini@unipg.it (A.D.); 11University of Turin, Department of Oncology, Division of Surgical Oncology and Digestive Surgery, San Luigi University Hospital, Orbassano, 10043 Torino, Italy; rossella.reddavid@unito.it; 12Department of Digestive Surgery, A. Gemelli Hospital, Catholic University, 00168 Roma, Italy; fausto.rosa@policlinicogemelli.it; 13Digestive Surgery Unit, European Institute of Oncology IRCCS, 20100 Milano, Italy; uberto.fumagalliromario@ieo.it

**Keywords:** gastric cancer, metastases, multimodal treatment, curative surgery, chemotherapy

## Abstract

**Simple Summary:**

The Italian Research Group on Gastric Cancer (GIRCG) developed a prospective registry called Meta-Gastro to collect data about stage IV gastric cancer. Data from Meta-Gastro contribute to the debate on oligometastatic gastric cancer definition, looking for the presence of prognostic factors in the metastatic population.

**Abstract:**

Background: Interest in the field of metastatic gastric cancer has grown in recent years, and the identification of oligometastatic patients plays a critical role as it consents to their inclusion in multimodal treatment strategies, which include systemic therapy but also surgery with curative intent. To collect sound clinical data on this subject, The Italian Research Group on Gastric Cancer developed a prospective multicentric observational register of metastatic gastric cancer patients called META-GASTRO. Methods: Data on 383 patients in Meta-Gastro were mined to help our understanding of oligometastatic, according to its double definition: quantitative/anatomical and dynamic. Results: the quantitative/anatomical definition applies to single-site metastases independently from the metastatic site (*p* < 0.001) to peritoneal metastases with PCI ≤ 12 (*p* = 0.009), to 1 or 2 hepatic metastases (*p* = 0.024) and nodal metastases in station 16 (*p* = 0.002). The dynamic definition applies to a percentage of cases variable according to the metastatic site: 8%, 13.5 and 23.8% for hepatic, lymphatic and peritoneal sites, respectively. In all cases, the OS of patients benefitting from conversion therapy was similar to those of cases deemed operable at diagnosis and operated after neoadjuvant chemotherapy. Conclusions: META-GASTRO supports the two-fold definition of oligometastatic gastric cancer: the quantitative/anatomical one, which accounts for 30% of our population, and the dynamic one, observed in 16% of our cases.

## 1. Introduction

During the last two decades, the Italian Research Group on Gastric Cancer (GIRCG) has been deeply involved in the clinical research concerning stage IV Gastric Cancer. While studying our cohorts [1], we fully understood the limitation of retrospective analysis and appreciated the self-imposing need for a more reliable instrument capable of originating high-quality clinical data for the progression of knowledge.

During the same timeframe, the cultural approach to the disease evolved in a remarkable way with the proposal of new classifications of the metastatic stage [2], the concept of “oligometastic” [3,4,5] and the introduction of important novelties in the oncologic approach, driven, among others, by tumor molecular biology [6].

After a long incubational process, a prospective observational register of metastatic gastric cancer patients called META-GASTRO was started in 2018. It enrolls all stage IV gastric cancer patients observed in the participating institutions.

With this publication originated by the META-GASTRO register, we would like to participate with real-life clinical data to the debate concerning the definition of oligometastatic gastric cancer. This debate is gaining popularity, and the available literature records some meaningful differences in this definition; for the first time, these will be challenged by clinical evidence originating from a Western cohort.

## 2. Materials and Methods

META-GASTRO is a prospective observational register of stage IV gastric cancer patients observed and treated in the different participating institutions. It was promoted by the Italian Research Group on Gastric Cancer with the intent to collect the largest number of metastatic patients, whose inclusion is based on the simplest criteria: the presence of metastases from gastric cancer. The inclusion in the study was proposed to all metastatic patients aged ≥18 years old; the sole exclusion criteria was the patient’s denial to participate. Discussion of each case at diagnosis and during the therapeutic course at the Multidisciplinary Tumour Board (MTB) is mandatory for inclusion in the register. One of the preeminent characteristics of META-GASTRO is the standardized revision of all clinical and radiological data by a dedicated multi-institutional team to guarantee the highest homogeneity within the different categories of data. META-Gastro received final approval in 2018 by the Ethic Committee (Comitato Etico Regionale per la Sperimentazione Clinica della Regione Toscana; Protocol 13082_2018; issued 20-04-2018); it is participated by 12 referring Italian Institutions. In the present paper, we are reporting data from patients included in the registry from September 2018 to September 2022. Although this register was originated and sponsored by the GIRCG surgical community, it is also supported by dedicated radiologists, oncologists and pathologists who participate in the GIRCG scientific activity. Patients are registered in a prospective manner, and data are progressively recorded following the patients’ clinical course. The database consists of 167 different fields reporting clinical, endoscopic, pathologic and imaging data at diagnosis, those emerging from surgical exploration if performed, and those concerning the specific oncologic treatment; clinical and instrumental follow-up information are also reported. The META-GASTRO register pursues the progress of clinical practice originated by big-data analysis.

The META-GASTRO database was investigated to define the concept of oligometastasis in gastric cancer with the support of the best clinical data available today.

### 2.1. Quantitative or Anatomical and Dynamic Definition of “Oligometastatic Gastric Cancer”

Adhering to the term’s etymology, the quantitative or anatomical definition of oligometastatic gastric cancer was investigated with multiple approaches. First, we verified the prognostic impact of the different metastatic sites (hepatic, peritoneal, lymph nodal, hematogenous extra-hepatic) and their association; subsequently, we investigated the prognostic impact of the different biologic categories proposed by Yoshida et al. [2]. Finally, we studied the cut-off position of the metastatic burden, considering the different metastatic sites by means of quantitative or anatomical measures. The peritoneal metastatic bulk was stratified according to the “P” classification proposed by the Japanese Gastric Cancer Association and the Peritoneal Cancer Index (PCI). In the latter case, the cut-offs of 6 and 12 were chosen according to the literature [7,8,9]. The hepatic metastatic bulk was expressed as the number of metastases with a cut-off at 2 and 5. Finally, the lymphatic metastatic bulk received an anatomical description and was evaluated according to the involved nodal station, from station 12p upwards; when multiple lymphatic stations resulted pathologic, the furthest was considered.

Following the literature guide [2,3,4,10,11], we also explored the dynamic definition of oligometastatic GC. Thus, we investigated the effect of oncologic treatments in terms of down-staging from a non-resectable status defined at the first MTB discussion to the resectable status, always referring to the different metastatic sites. Furthermore, we explored the survival performance of these patients. Patients submitted to surgical treatment were divided into 2 groups: those considered technically operable with radical intent (both on primary and on metastases) at diagnosis and those judged non-resectable (for technical or oncological reasons). At re-evaluation after neoadjuvant chemotherapy, patients in the first group (superimposable to Yoshida’s category 1) were submitted to surgery with curative intent on primary and metastases. Similarly, after response to intensive chemotherapy, patients in group 2 (scattered through Yoshida’s categories 2, 3 and 4) were submitted to surgery with curative intent on primary and metastases; they formed the conversion therapy group.

### 2.2. Endpoints of the Study

For the identification of the quantitative/anatomical definition of oligometastatic, we performed a thorough evaluation of survival of stage IV gastric cancer patients according to the site of metastases and burden of disease. For the dynamic definition, we evaluated survival according to the response to the different therapeutic strategies.

### 2.3. Statistical Analysis

Continuous variables are presented as median ± standard error and confidence interval or interquartile range (IQR, 25–75%). Statistical significance was rated at *p* < 0.050. Overall survival (OS) was measured from the date of diagnosis to the date of death or latest follow-up. Survival curves were generated by the Kaplan–Meier method and compared using the log-rank test. Survival rates are presented as percentages and a 95% confidence interval. Variables that resulted statistically significant (*p* < 0.050) at univariate analysis were considered for multivariate analysis with the Cox proportional hazards model. Statistical analysis was performed using SPSS software, package 18.

## 3. Results

The METAGASTRO registry currently collects certified data on 383 patients whose median age is 69 years (59–75); all observed patients were included in the register as we did not receive a single denial of the proposal. Besides endoscopy, the diagnostic work-up included at least a CT scan in all cases. Table 1 reports the main clinical data of our population. Diagnostic laparoscopy was employed in 130 patients, and 26 received an explorative laparotomy. A total of 13 patients received upfront surgery for palliation of acute symptoms such as perforation, bleeding and obstruction; 35 received the best supportive treatment; and 335 patients received a dedicated elective treatment of their disease, which consisted of chemotherapy, associated in 111 cases to surgical procedures. The median OS of our cohort was 14.3 months (12.6–15.9). After discharge, all patients were submitted to a regular follow-up, regulated by the different therapeutic options; the median follow-up time for the entire cohort is 9.2 months.

### 3.1. Quantitative or Anatomical Definition

#### 3.1.1. Metastatic Sites

A single metastatic site was observed at diagnosis in 250 patients, while in 133 cases, multiple metastatic sites were detected. Considering those presenting a single site, this was peritoneal in 125, hepatic in 62, lymph nodal in 58 and hematogenous beyond the hepatic filter in five cases; this latter group was not further investigated due to the small number of cases. Median overall survival ranged from 21.6 months (13.9–29.3) for patients with lymph node metastases to 16.7 (12.2–21.2) and 16.2 months (12.9–19.5) for those with hepatic and peritoneal metastases, respectively. Overall survival curves of these different groups of patients are reported in Figure 1a and show similar behavior without relevant differences between them. On the contrary, the 133 patients presenting at diagnosis more than one metastatic site had a median survival of 10.5 months, and their survival curve showed a poorer performance if compared to the single site counterparts (*p* < 0.001).

#### 3.1.2. Biologic Categories Proposed by Yoshida et al. [2]

The stratification of our cohort according to the biological categories proposed by Yoshida et al. [2] recorded 44 patients in category 1 and 131 in category 2, while 108 and 100 patients were enrolled in category 3 and 4, respectively. Median overall survival ranged from 21.5 months (18.5–24.4) for category 1 to 15.5 (13.3–17.6) and 16.1 (13.1–19.8) months for categories 2 and 3, respectively. Survival curves of these different groups of patients are reported in Figure 1b and show similar behavior without relevant differences between them. On the contrary, patients enrolled in category 4 had a median survival of 7.5 months (5.1–9.8), and their survival curve showed a poorer performance if compared to their counterparts (*p* < 0.001).

#### 3.1.3. Peritoneal Burden

Peritoneal metastases were detected in 214 patients. The peritoneum was the sole metastatic site in 125 cases; in 89 cases, peritoneal metastases were diagnosed in the context of a multiple-site metastatic disease.

Besides the 120 patients who received a surgical exploration, the radiologic Committee reached a consensus concerning the P stratification in 56 more patients; thus, a total of 177 patients was available for this analysis. Furthermore, we have a detailed description of the PCI in 93 cases, which were grouped with 14 cases presenting positive cytology at diagnostic laparoscopy but not macroscopic carcinosis.

Our data show (Figure 2a) that median survival was 15.4 (10.3–20.4), 11 (4.1–17.8) and 9.5 (5.2–13.8) months for P1a, P1b and P1c categories, respectively, *p* = n.s. According to the chosen PCI stratification (Figure 2b), our data show a median survival of 18.3 (14.9–21.6), 16.1 (12.4–19.7) and 12.4 (4.9–19.8) months for PCI < 6, 6–12 and >12, respectively; positive cytology was associated to the best median survival: 21.6 (5.2–37.9) months, *p* = 0.009.

Similar trends had been observed in the subgroups of patients presenting peritoneal involvement as a unique metastatic site.

#### 3.1.4. Hepatic Burden

Hepatic metastases were detected in 142 patients. The liver was the sole metastatic site in 62 cases, while in 80 cases, hepatic metastases were diagnosed in the context of a multiple-site metastatic disease.

Survival according to the hepatic metastatic burden, expressed as the number of metastases, is reported in Figure 3a. Median survival of patients affected by one or two hepatic metastases was 18.4 (6.8–29.9) months and compared favorably with the 12.1 (9.8–14.3) months of patients presenting three to five metastases, reaching statistical significance (*p* = 0.0015) when compared to the 8.4 (1.2–15.5) months of patients presenting scattered bilobar metastases. Similar data are observed in the subgroup of patients presenting hepatic involvement as a unique metastatic site.

#### 3.1.5. Lymphatic Burden

Extrapolation of data on lymphatic metastases was hampered by the centralized revision of all CT scans, which has not yet been completed as far as the anatomical location of nodal metastases is concerned. We have a complete description of the different metastatic stations in 67 patients; in 24 cases, this was the sole metastatic site.

Survival according to the different stations is reported in Figure 3b. Median survival of patients presenting metastases in station 16 was 30.2 (8.1–52.3) months and compared favorably with the 4.5 (0.0–9.1) months of patients presenting metastases in station 12p and 13, with the 8.4 (1.2–15.5) months of other distant abdominal stations, with the 15.5 (7.2–23.7) and 15.4 (14.3–16.4) months of patients presenting mediastinal or other distant extra-abdominal metastatic nodes, respectively (*p* = 0.002).

### 3.2. Dynamic Definition

#### 3.2.1. Peritoneal Metastases

After diagnostic work-up, 22 out of 214 patients (9.9%) presenting peritoneal metastases were considered technically resectable; among them, 7 had a double-site metastatic disease. They all received intensive chemotherapy, and 12 were subsequently operated on (primary and metastases); R0 was obtained in nine cases. These 22 patients represent the benchmark for survival evaluation. At diagnosis, peritoneal metastases were considered non-resectable in 192 cases: 18 received the best supportive treatment, 27 received palliative surgery (in 15 cases associated with CHT), and 147 received chemotherapy. At re-evaluation after intensive chemotherapy, 35 of the latter (23.8%) were deemed resectable and thus submitted to surgical resection of both primary and peritoneal metastases. R0 was obtained in 18 cases.

The median survival of patients deemed operable at diagnosis and of those receiving conversion therapy was 30.2 (15.4–44.9) months and 22.2 (14.3–30.1) months, respectively. These compared favorably (*p* < 0.001) with the median survival of patients who received CHT alone or palliative surgery: 11.2 (6.2–15.2) months and 8.2 (1.1–8.8) months, respectively, as shown in Figure 4a.

#### 3.2.2. Hepatic Metastases

After diagnostic work-up, 22 out of 142 patients (15.5%) presenting hepatic metastases were considered technically resectable. They all received intensive chemotherapy, and 12 were subsequently operated on (R0 was obtained in 11 cases). These 22 patients represent the benchmark for survival evaluation. At diagnosis, hepatic metastases were considered non-resectable in 120 cases: 8 received the best supportive treatment, 13 received palliative surgery (in 3 cases associated with CHT), and 99 received chemotherapy. At re-evaluation after intensive chemotherapy, eight of the latter (8%) were deemed resectable and thus submitted to surgical resection of both primary and hepatic metastases (R0 was obtained in three cases).

The median survival of patients deemed operable at diagnosis and of those receiving conversion therapy was 14.8 (10–19.5) months and 27.6 (23.4–31.7) months, respectively. These compared favorably (*p*-0.001) with the median survival of patients who received palliative surgery or CHT alone: 13.4 (0.3–26.4) months and 10.9 (8.6–13.1) months, respectively, as shown in Figure 4b.

#### 3.2.3. Lymph Nodes Metastases

After diagnostic work-up, 40 out of 150 patients (26.6%) presenting distant nodal metastases were considered technically resectable; among them, 4 had a double-site metastatic disease. They all received intensive chemotherapy, and 26 were subsequently operated on (primary and metastases); R0 was obtained in 21 cases. These 40 patients represent the benchmark for survival evaluation. At diagnosis, lymph nodal metastases were considered non-resectable in 110 cases: 5 received the best supportive treatment, 9 received palliative surgery, and 96 received chemotherapy. At re-evaluation, after intensive chemotherapy, 13 of the latter (13.5%) were deemed resectable and thus submitted to surgical resection of both primary and nodal metastases. R0 was obtained in six cases.

The median survival of patients deemed operable at diagnosis and of those receiving conversion therapy was 30.2 (21.1–39.2) months and 22.5 (15.1–29.3) months, respectively. These compared favorably (*p* < 0.001) with the median survival of patients who received palliative surgery or CHT alone: 8.2 (0.1–16.3) months and 11 (8.8–13.1) months, respectively, as shown in Figure 4c.

#### 3.2.4. Survival According to Treatment Received

Considering the entire cohort, our data show (Figure 5) that median survival after CHT and curative surgery for technically resectable metastatic gastric cancer at diagnosis was 30.2 (19.8–40.5) months and 25.4 (19.1–31.8) months after conversion surgery for metastases deemed non-resectable at diagnosis. In the absence of effective surgical removal of the tumor burden, survival drops to 13.4 (3.3–23.5) and 10.9 (9.0–12.7) months for patients receiving non-curative surgery or CHT alone. Table 2 reports surgical details.

## 4. Discussion

The most important data emerging from the META-GASTRO database is the demonstration that in real-life clinical practice, the clinical entity defined as “oligometastatic gastric cancer” exists and accounts for the non-negligible percentage of 45% of our population.

Oligometastatic gastric cancer has a two-fold definition, reflecting different ways to measure it. In the etymological definition, oligometastatic reflects a quantitative or, in a broader sense, anatomical measure directly correlated to the metastatic bulk. This definition has a relevant clinical value since it is readily available and thus capable of driving clinical strategies toward an aggressive approach to the disease. The second definition applies to the restricted subgroup of patients submitted to intensive chemotherapy, as it measures the effect of chemotherapy in terms of down-staging from non-operable (for oncological or technical reasons) to operable condition; in this case, it is appropriate to define the oligometastatic status as dynamic or acquired. In the clinical approach to the metastatic gastric cancer patient, both the quantitative and the dynamic definitions must be considered.

Considering the anatomical or quantitative definition, the attention is at first attracted by the different prognostic roles displayed by single-site or multiple-site metastatic burdens; these data are confirmed by the cohort stratification according to the different categories proposed by Yoshida et al. Upon this basis, “oligometastatic” apparently correlates to a single metastatic site. Moreover, the present prospectively collected data confirm those highlighted by our previous retrospective work [1], demonstrating the similar prognostic role displayed by the different metastatic sites when they represent the unique location of the metastatic burden. This is a relevant issue as it implies that all the possible metastatic localizations should be addressed with the same proactive attitude to identify those patients who display the best survival chances and thus deserve the optimization of the oncologic treatment. A rapid glance at the literature suggests that a certain awareness is appreciable upon hepatic metastases [12,13,14], but it is not the same when it comes to the peritoneal [15] and lymphatic spread of the disease. Based on our data, single-site metastases appear as a prerequisite or, better, a facilitating condition in the quest for the oligometastatic patient, which deserves more detailed evaluations within each different metastatic site.

For this reason, we explored the metastatic bulk in the three most common metastatic sites. Despite limitations arising from the relatively small number of patients in each cohort, META-GASTRO offered interesting and unexpected insights. Considering peritoneal metastases, the “P” classification proposed by the JGCA did not correlate with survival. This result does not support the recommendations issued by the JGCA, which consider the P1 (or P1a) category oligometastatic. On the contrary, a correlation exists between median survival and PCI, and a PCI cut-off positioned at 12 seems capable of stratifying patients into best and worst survivors. At present, we can consider patients presenting a PCI < 12 oligometastatic; further development of our registry will be able to provide a more accurate stratification of risk according to lower PCI values, if any exist. In parallel, we must highlight the best survival performance displayed by patients presenting positive cytology at peritoneal lavage. If these data are confirmed on a larger scale, in full accordance with Yoshida et al. and the JGCA guidelines, oligometastatic will subtly apply to patients at high risk of but not affected by peritoneal metastases, confirming the prognostic role of peritoneal cytology. We must warn here that these results must be handled with care because they need confirmation. In fact, they had been extrapolated from relatively small cohorts; for this reason and for the results themselves, which were partially unexpected, this metastatic site will be studied in depth as soon as Meta-Gastro increases the recruitment.

Regarding the hepatic burden of the disease, our data, in accordance with the literature, show a neat inverse correlation between the number of metastases and prognosis. This element has a relevant clinical and prognostic value since it is easily available in the clinical setting; for this reason, its confirmation from a prospective analysis displays a critical value. In our cohort, positioning the cut-off number of metastases at two and five for low and intermediate-risk categories, respectively, allowed the appreciation of different median survival among groups. In full agreement with the prevailing retrospective literature, our data support the definition of oligometastatic for those cases presenting one or two metastases into the hepatic parenchyma; the progressive increase in the numerosity in our register will be able to perform more accurate evaluations and include the lesion’s diameter as additional quantitative criteria.

Regarding lymphatic metastases, our data suggest that oligometastasis should be declined in an anatomical rather than quantitative dimension. At present, we showed that para-aortic lymph node metastases have a more favorable prognostic value, and thus, metastasis in this location should be considered oligometastatic. Furthermore, posterior nodes located in stations 12p and 13 suffer a poor prognosis, as already signaled by our group [16]. We would have been able to further stratify station 16 in its different substations, but this was impossible due to the low number of cases.

A point in the above analysis deserves particular attention: in our stratifications, patients had been grouped according to their metastatic bulk, independent of the treatment they received. This introduces a bias in the interpretation of survival curves since patients presenting more favorable conditions are more likely to receive aggressive multimodal treatments, including surgery with curative intent, both on gastric cancer and metastases. Among patients considered at diagnosis “oligometastatic” according to the quantitative or anatomical criteria, this was observed in a percentage ranging from 54.5% to 65%, depending on the metastatic site, while non-oligometastatic patients did not receive surgery with curative intent. This observation introduces the second hot topic (besides the definition itself) of our discussion: the role of the different therapeutic strategies, which may dramatically affect survival performances and thus interfere with the definition of oligometastatic, which is intrinsically correlated to a better oncologic outcome.

This tight interconnection is at the basis of the dynamic definition of oligometastatic gastric cancer, which is measured in terms of downstaging from inoperable to operable conditions and conceived in view of potentially curative surgery, both on the primary tumor and on the metastases. In our experience, all these patients received full conversion therapy, completed by conversion surgery with curative intent. META-GASTRO suggests that the dynamic form of oligometastatic may have different penetrations in the different metastatic sites, ranging from 8% for hepatic metastases to 24% for peritoneal metastases. However, we note that survival curves of the “conversion” groups appeared superimposable to those of patients considered oligometastatic at diagnosis, reproducing on Western population data observed in Eastern countries such as Japan, Korea, and China. From this point of view, META-GASTRO confirms the importance of curative surgical ablation of the neoplastic bulk, which demonstrated an independent prognostic value in our previous retrospective works. Once again, our data show that in referral centers that may guarantee common-sense guided surgical principles, major surgical morbidity has an acceptable rate of 15%, while mortality is maintained under 2%.

## 5. Limitation of the Study and Future Developments

This work has some limitations, mainly connected to the relatively small number of patients recruited. Even if, from a Western perspective, our population may appear huge, it is inadequate to allow all the necessary analyses through the different subsets of patients generated by our quest for oligometastatic gastric cancer. As already stated, data concerning peritoneal metastases need validation, while those referring to nodal metastases, with particular reference to station 16, need more in-depth studies. For this reason, we consider the present work as introductory and preliminary to subsequent analyses, which will be possible as soon as Meta-Gastro accrual becomes more robust. Notwithstanding, the quality of our data is extremely high due to the centralized multidisciplinary and multicentric revision of every single case; for this reason, we decided to contribute now to the scientific debate with real-life clinical data in order to support the big effort promoted by the entire oncological community aimed to the rescue and cure of at least a fraction of the less fortunate among our patients.

## 6. Conclusions

META-GASTRO supports a two-fold definition of oligometastatic gastric cancer detailing the quantitative/anatomic aspect of the disease, which accounts for 30% of our population, and the dynamic one, observed in 15% of our cases. In the case of single-site metastases, peritoneal involvement with PCI <12, one or two hepatic metastases and nodal metastases in station 16 should define oligometastatic gastric cancer according to the quantitative/anatomical definition. A good response to aggressive chemotherapy converting a non-operable status for technical or oncological reasons to a resectable status with curative intent both on primary and metastases defines dynamic oligometastatic gastric cancer. Curative surgical ablation of the entire neoplastic bulk represents a cornerstone for the possibility of cure and long survival.

## Figures and Tables

**Figure 1 cancers-16-00170-f001:**
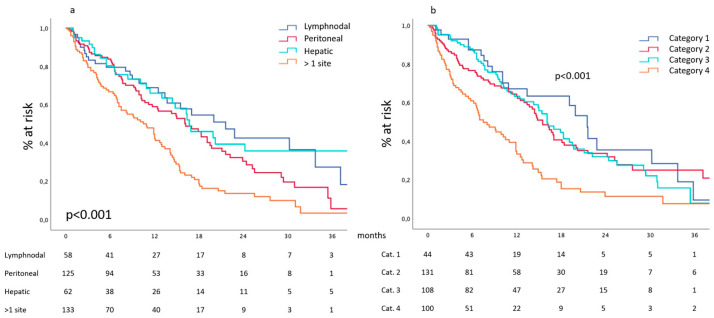
(**a**) Site-related survival and (**b**) survival according to the categories proposed by Yoshida et al.

**Figure 2 cancers-16-00170-f002:**
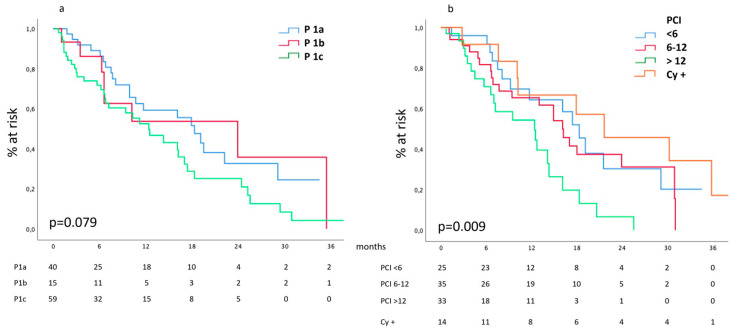
(**a**) Survival according to the Japanese P classification of peritoneal involvement and (**b**) to PCI and positive cytology.

**Figure 3 cancers-16-00170-f003:**
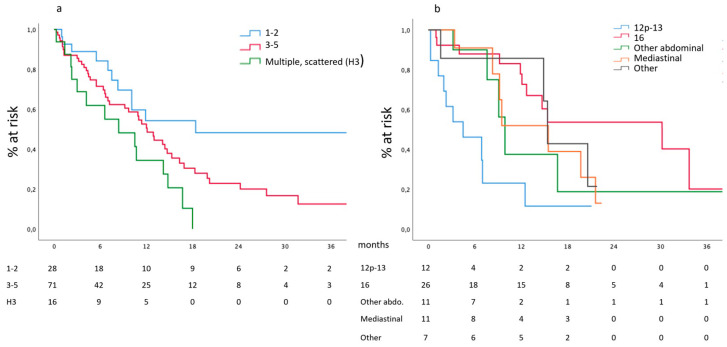
(**a**) Survival according to the number of hepatic metastases and (**b**) nodal involvement.

**Figure 4 cancers-16-00170-f004:**
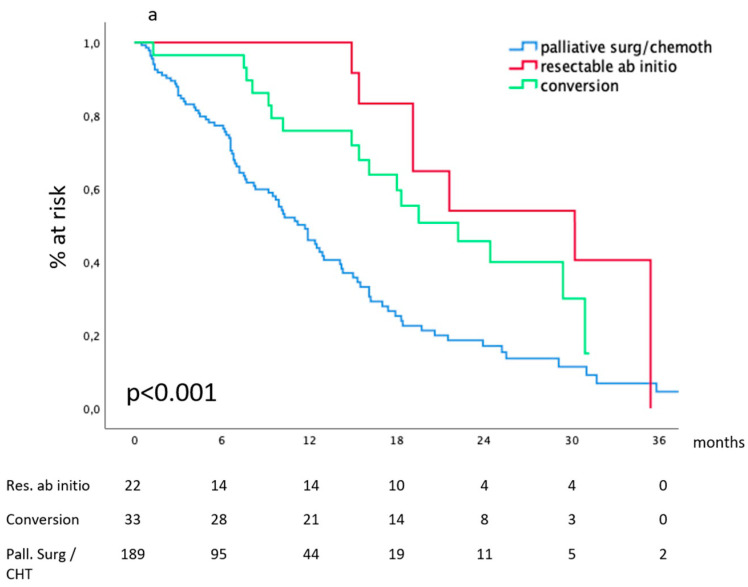
Survival according to the dynamic definition: (**a**) peritoneal metastases, (**b**) hepatic metastases, (**c**) lymph node metastases.

**Figure 5 cancers-16-00170-f005:**
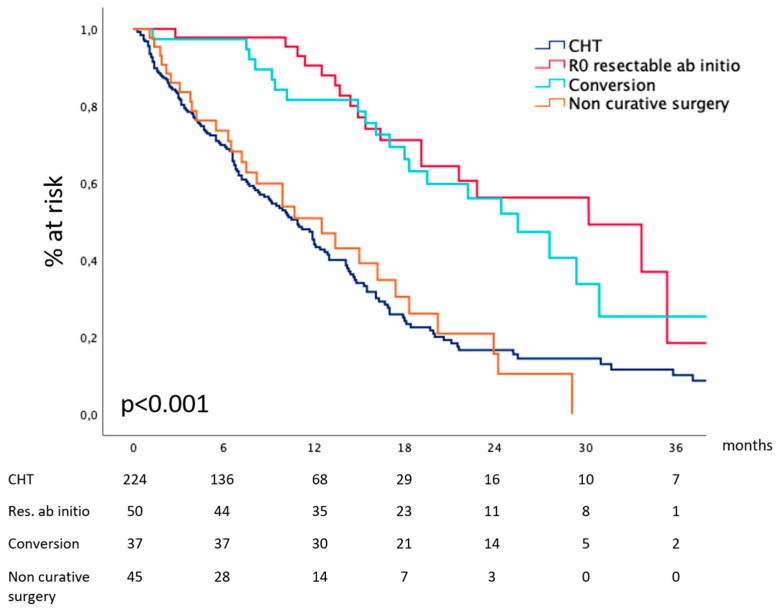
Survival according to the treatment received.

**Table 1 cancers-16-00170-t001:** Characteristics of the population.

		Nr.	%
Tumor Site			
	Siewert 3	11	2.9
	Proximal	148	38.6
	Intermediate	59	15.4
	Distal	124	32.4
	Linitis	28	7.3
	Gastric stump	13	3.4
cT ^1^			
	1–2	11	2.9
	3–4	231	60.3
	x ^3^	141	36.8
cN ^2^			
	0	23	6.0
	1	285	74.4
	x ^3^	75	19.6
Histology			
	Adenocarcinoma NOS	164	42.8
	Diffuse/poorly cohesive/signet ring cell	129	33.7
	Intestinal/tubular	74	19.3
	Mixed	6	1.6
	Other	10	2.6
Cytology			
	Not performed	251	65.5
	Positive	48	12.6
	Negative	84	21.9
ECOG ^4^			
	0–1	187	48.8
	2	127	33.2
	3–4	69	18.0
Chemotherapy			
	Fluorouracil + platinoids	176	52.7
	Fluorouracil + taxane + platinoids	75	22.4
	Trastuzumab ± others	25	7.5
	Other	59	16.6

^1–2^ cT-cN: clincal TNM classification, 8th edition; ^3^ x: unknown; ^4^ ECOG: European Cooperative Oncology Group performance status scale.

**Table 2 cancers-16-00170-t002:** Details on surgery and its complications.

		Nr.	%
Type of surgery			
	Palliative resection	29	22.0
	Resectable ab initio	49	37.1
	Conversion surgery	38	28.8
	Non-resective surgical palliation	16	12.1
Curativity of surgery			
	R0	64	48.5
	R1	16	12.1
	R2	42	31.8
	Rc0 ^1^	10	7.6
Post-op. complication (Clavien-Dindo)			
	0-1-2	109	82.6
	3a	12	9.1
	3b	6	4.5
	4	3	2.3
	5	2	1.5

^1^ Rc0: complete clinical regression of metastases.

## Data Availability

The data presented in this study are available on request from the corresponding author. The data are not publicly available due to privacy and ethical reasons.

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
