# Peer review of "Oligometastatic Gastric Cancer: Clinical Data from the Meta-Gastro Prospective Register of the Italian Research Group on Gastric Cancer"

_cancers, 2023, doi:10.3390/cancers16010170_

Round 1

Reviewer 1 Report

Comments and Suggestions for Authors

Comment.

I commend you on the meticulous work presented in your manuscript, " Oligometastic Gastric Cancer: Clinical Data from the 3 Meta Gastro Prospective Register of the Italian Research Group on Gastric Cancer." Your dedication to investigating the nuances of oligometastatic gastric cancer is evident, and utilization of the META-GASTRO database provides a valuable real-life clinical perspective. While your study contributes significantly to our understanding of oligometastatic gastric cancer, I believe addressing a few major and minor points will further strengthen the impact and clarity of your work.

Major Points:

Definition of oligometastatic Gastric Cancer

The definition of oligometastasis presented in your study is not your original, but a validation of the definitions used in other studies. The authors presented outcomes for the patients included in this study, but there is insufficient validation of these definitions of Oligometastatic Gastric Cancer. This study lacks a clear explanation of the methodology as a prospective study, including patient eligibility criteria. In addition, this study had the possibility of selection bias in prospective studies.

Oligomatastasis of Peritoneal Dissemination

I think that it is difficult to agree with the definition of oligometastasis of peritoneal dissemination in your study. According to Yoshida’s classification, only CY1 is an oligometa, and I think so. According to the JGCA treatment guidelines, P1 (P1a) and a small amount of P1b (P2) may be considered oligometastatic; however, PCI≤12 seems too widespread to be oligometa. This definition is difficult to apply clinically, because it cannot be evaluated using imaging studies. Furthermore, the reference literature used for this definition is based on cytoreductive surgical cases, which may have a different treatment approach from systemic chemotherapy.

Oligomatastasis of LN16 Metastasis.

LN16 metastases are qualitatively and quantitatively diverse. More information and clarification is needed regarding the criteria for LN16 oligomata. I believe that the lymph node meta limited to LN16a2/b1 would be oligometa.

Inclusion criteria and informed consent.

The eligibility and exclusion criteria for participation in this prospective study should be described. There appeared to be no mention of patient informed consent or age restrictions. These statements are essential.

Handling of BSC and upfront surgery

Patients treated with BSC or upfront surgery should be excluded.

Localization of Tumors:

The tumor location should be described. EGJ cases should be excluded if they were included.

Surgical Indications:

A detailed explanation of the surgical indications, especially for initially resectable tumors (induction chemotherapy) and conversion surgery, is essential.

Sample Size Limitation Clarification.

The acknowledgment of the relatively small sample size is good, but a more in-depth discussion of how this limitation might impact the study's conclusions is crucial. This enhances the reader’s understanding.

Specificity of Future Research Directions

More specific guidelines should be provided when discussing future research. It is necessary to discuss how your results will affect future treatment strategies. This will provide a clear direction for researchers and readers.

Minor Points:

Lines 56-60;

This section is unrelated to the present study. This could be taken as a self-citation; therefore, I recommend deleting this sentence.

Lines 74-82, 139-153:

The journal template remains in place. I recommend deleting this sentence.

Line 95-96:

The follow-up period for the patients who underwent surgery was relatively short. You need to examine in more detail the possible impact of this short follow-up period on the evaluation of survival outcomes.

Figures:

The figures are difficult to see. Please correct these so that it is easy to see. The pixels are extremely rough.

Author Response

Reviewer 1.

We appreciate the observations raised by the Reviewer and we are grateful for the precious notes and suggestions upon which we hope we’ve been able to improve the quality of our work.

Herein, please find a point by point response to Reviewer’s comment.

Major points:

  1. Definition of oligometastatic gastric cancer

Reviewer: The definition of oligometastasis presented in your study is not your original, but a validation of the definitions used in other studies. The authors presented outcomes for the patients included in this study, but there is insufficient validation of these definitions of Oligometastatic Gastric Cancer. This study lacks a clear explanation of the methodology as a prospective study, including patient eligibility criteria. In addition, this study had the possibility of selection bias in prospective studies.

Answer: Thank you for this comment upon which we fully agree. There are several definitions of oligometastasis and our aim is to contribute to define this topic. Definition of oligometastatic Gastric Cancer proposed in the literature (OMEC, FLOT-3 and those from the Far East) present several differences. Our aim is to contribute to the discussion. We fully agree that the definition of oligometastatic we referred to had been extrapolated from the literature. We have made this point clearer in the introduction (red text) and expanded the references, previously scattered through the text.

We also agree with the fact that validation of oligometastatic definition, at least in the West, is still lacking and that the existing papers have a speculative nature since they arise from expert opinion and consensus reached by delphy methodology. As stated in the introduction, the idea of a prospective observational database arose to verify upon real-life clinical data the proposed definitions.

As for methodology, Meta-gastro is a prospective observational register whose aim is to pursue the progress of clinical practice through big-data analysis. We made all efforts to achieve the highest level of data quality through the centralized revision of all data, including cross-sectional imaging. This methodology was clearly stated trough the introduction and materials & methods (in particular along lines 77-88 and 94-101); further explanations concerning data mining and stratification criteria have been detailed in paragraph 2.1 (red text) and 2.2 (new paragraph)

Concerning the last point, we are unable to detect selection biases since all observed patients have been included in the study. We have reported this in the Result section, lines 159-160. From our perspective this is a strong hold of our register which includes all metastatic patients including those submitted to upfront surgery due to symptomatic disease and those who could only receive supportive care. When considering the metastatic landscape all these must be included to offer a complete representation of this pathology.

  1. Oligomatastasis of Peritoneal Dissemination

Reviewer: I think that it is difficult to agree with the definition of oligometastasis of peritoneal dissemination in your study. According to Yoshida’s classification, only CY1 is an oligometa, and I think so. According to the JGCA treatment guidelines, P1 (P1a) and a small amount of P1b (P2) may be considered oligometastatic; however, PCI≤12 seems too widespread to be oligometa. This definition is difficult to apply clinically, because it cannot be evaluated using imaging studies. Furthermore, the reference literature used for this definition is based on cytoreductive surgical cases, which may have a different treatment approach from systemic chemotherapy.

Answer. We are grateful for this note which pinpoints one of the most important issue raised by our results. As per peritoneal metastases, the definition of oligometastatic is intriguing. We completely agree with your vision: we felt that a PCI <6 could be the sole possibility to define oligometastatic peritoneal involvement. Results achieved by patients in the 6-12 PCI cohort were unexpected and surprising. The scant data from literature, the different expert opinions upon this topic as well as the recommendations or suggestions originated by the existing guidelines aren’t supported by clinical evidence originated from western cohorts. As often happens in the clinical research, real life data such as those from the Meta-Gastro may provide precious clinical support to our believes but also challenge our vision. This latter was the case for PCI. 

Pragmatically, answering your questions:

- CY1 were the best performers; they had been included into the oligometastatic category, accordingly to the Reviewer’s and Yoshida’s vision. This was stressed in the discussion (lines 330-331, in red);

- the impact of the Japanese P classification was evaluated by means of 120 patients who underwent surgical exploration and of 56 non-operated patients for whom the radiologic committee reached a consensus upon the P stratification (P1-3 or P1a-c). Kaplan-Meyer survival curves originated from these patients do not permit any prognostic differentiation. It must be noted that the stratification had been carried on upon data at diagnosis, irrespective to treatment strategies. We highlighted this discrepancy in discussion (lines 323-324); furthermore, we stressed that all results achieved on peritoneal metastases must be handled with care (lines 333-337)

-  The stratification according to the PCI was performed on data from 107 cases. Of them, 14 resulted CY1 without peritoneal metastases and 93 had a detailed description of the peritoneal burden of disease including PCI calculation. This information was achieved by surgical means, more frequently by staging laparoscopy. The cut-off values had been extrapolated from expert opinions and from the existing literature. We referred to cytoreductive series because these benefit of the best definition of the peritoneal burden and because there are no alternatives. As expected, we observed a good survival performance for PCI<6, irrespective of the treatment strategy: oligometastatic! We were surprised by a similar performance for PCI in the 6-12 range. This is the most important challenge exerted by Meta-Gastro on our vision. However, if we can’t scotomize this result, we also don’t handle it as a statement. We made this point more intelligible in the discussion, adding a warning to handle with care these results, extrapolated from relatively small cohorts (lines 333-337).

  1. Oligomatastasis of LN16 Metastasis.

Reviewer: LN16 metastases are qualitatively and quantitatively diverse. More information and clarification is needed regarding the criteria for LN16 oligomata. I believe that the lymph node meta limited to LN16a2/b1 would be oligometa.

Answer: We agree with the Reviewer and would have had the possibility to investigate the prognostic value of each different subtype of station 16 metastases. Also in this case, however, the number of patients did not allow this stratification. It will be object of further studies. We noted this point in discussion (lines 357-358).

  1. Inclusion criteria and informed consent.

Reviewer: The eligibility and exclusion criteria for participation in this prospective study should be described. There appeared to be no mention of patient informed consent or age restrictions. These statements are essential.

Answer: We agree to the concern raised about eligibility and exclusion criteria, thus we added a sentence in material & methods (lines 81-82),

As far as the patient’s consent it had been declared at the end of the manuscript in a dedicated form, according to Editorial Instructions.

  1. Handling of BSC and upfront surgery

Reviewer: Patients treated with BSC or upfront surgery should be excluded.

Answer: we do not agree upon this point. In order to faithfully reproduce the clinical landscape of metastatic gastric cancer in Italy and to avoid any selection bias, we decided to include in Meta-Gastro all metastatic patients without any exclusion, in fact they exist, and an observational study can not erase them. BST patients generally presented multiple metastatic sites and no one among them could have been categorized as oligometastatic from a quantitative/anatomic point of view; their performance must be considered when analysing the different metastatic categories but, being excluded from treatment-related analyses, they do not hamper the appreciation of the dynamic definition of oligometastatic. At opposite, patients submitted to upfront surgery for the need to palliate symptoms merit our consideration: beside palliation, some of them benefitted of cytoreduction and the majority received CHT in the post-operative period. In our perspective they will form a separate study group when and if numbers will allow for this.

  1. Localization of Tumors:

Reviewer: The tumor location should be described. EGJ cases should be excluded if they were included.

Answer: Meta-Gastro enrols patients affected by gastric cancer. In regard to proximal tumors only Siewert 3 are included. Tumor location was inserted in Table 1 (red text).

  1. Surgical Indications:

Reviewer: A detailed explanation of the surgical indications, especially for initially resectable tumors (induction chemotherapy) and conversion surgery, is essential.

Answer: Thank-you for this note. We added the information in material & methods, lines 130-138.

  1. Sample Size Limitation Clarification.

Reviewer: The acknowledgment of the relatively small sample size is good, but a more in-depth discussion of how this limitation might impact the study's conclusions is crucial. This enhances the reader’s understanding.

Answer: good observation. We developed this point in discussion: lines 385-391.

Minor points:

  1. Lines 56-60;

Reviewer: This section is unrelated to the present study. This could be taken as a self-citation; therefore, I recommend deleting this sentence.

Answer:  We agree with this vision, thus the sentence was deleted and bibliography was revised accordingly.

  1. Line 74-82, 139-153:

Reviewer: The journal template remains in place. I recommend deleting this sentence.

Answer:  We agree with this note. It was an editing error.

  1. Line 95-96:

Reviewer: The follow-up period for the patients who underwent surgery was relatively short. You need to examine in more detail the possible impact of this short follow-up period on the evaluation of survival outcomes.

Answer: two answers to this point. First: This sentence was placed in the wrong place. It has been relocated in the “Results” section, lines 163-165. Second: the median follow-up refers to the entire cohort and not to the sole group of patients who could benefit of curative intent treatments such as conversion surgery or surgery after neoadjuvant CHT. Unfortunately, as commented in discussion, these patients do not represent the majority of cases.

  1. Figures:

Reviewer: The figures are difficult to see. Please correct these so that it is easy to see. The pixels are extremely rough.

Answer: You are right. We changed the numeration and made figures bigger and of better quality.

Reviewer 2 Report

Comments and Suggestions for Authors

Referee Report

This study investigates oligometastatic gastric cancer using clinical data from Italy's meta-gastro prospective register in a prospective observational manner. The work is comprehensive and respectable. However, I have a few minor concerns:

  1. Author Affiliation: This work lists two first authors. Typically, only one first author is accepted by journals. I understand the reasoning behind this arrangement based on the authors’ contributions.
  2. Title: The title might benefit from rephrasing, as "Prospective Observational Study" seems separate from the rest of the title. The authors could aim to unify the wording.
  3. Abstract: The abstract lacks a clear indication of the primary aim of the manuscript. It's unclear if the goal is to conduct a prospective observational study on oligometastatic gastric cancer or to establish a meta-gastro prospective register for such observational studies. Clarity regarding the aim in the Abstract would be beneficial.
  4. Introduction: While the Introduction briefly covers the background of oligometastatic gastric cancer and some statistics, it could be expanded to provide a more detailed context.
  5. Methods and Materials: Information regarding the number of patients involved in the register is missing. Details regarding data filtration and mining processes could be included.
  6. Section 2.2: It would be beneficial if the authors specified the software and version used for statistical and error analyses.
  7. Figures 1-3: These figures appear too small for clear viewing. Enlarging them, if feasible, would improve readability.
  8. Figure 2: Consider combining Figure 2(a,b) with Figure 2(c,d) or labeling Figure 2(c,d) as a separate figure. Note that the resolution of Figure 2(c,d) is lower than Figure 2(a,b).
  9. Duplicate Figure 3: The manuscript contains two Figure 3s, which needs clarification or correction.
  10. Conclusion: Mentioning future work related to this study in the conclusion would be beneficial.
Comments on the Quality of English Language

No problem to read the paper.

Author Response

Reviewer 2

We appreciate the observations raised by the Reviewer and we are grateful for the precious notes and suggestions upon which we hope we’ve been able to improve the quality of our work.

Herein, please find a point-by-point response to Reviewer’s comment.

Reviewer: Author affiliation: this work lists 2 first authors. Typically, only 1 first Author is accepted by journals. I understand the reasoning behind this arrangement based on the Author’s contributions.

Answer: We appreciate the Reviewer’s understanding of this point. Leadership in multi-institutional works is often shared and co-participated. In particular, Meta-Gastro required a tremendous effort from its conception through all its development stages. In all circumstances our 2 first Authors exerted their leading role. Our desire was to recognize their effort. This was also guaranteed by Cancers Editorial strategy which does not contraindicate a double first authorship.

Reviewer: Title: the title might benefit from rephrasing, as “prospective observational study” seems separate from the rest of the title. The authors could aim to unify the wording.

Answer: the phrase “Prospective observational study” is not part of the title. It had been inserted by the Editorial Committee, probably to better define the category of the contribution.

Reviewer: Abstract: the abstract lacks a clear indication of the primary aim of the manuscript. It’s unclear if the goal is to conduct a prospective observational study on oligometastatic gastric cancer or to establish a meta-gastro prospective register for such observational studies. Clarity regarding the aim in the abstract would be beneficial.

Answer: Thank-you. The abstract was re-written considering your suggestion.

Reviewer: Introduction: While the Introduction briefly covers the background of oligometastatic gastric cancer and some statistics, it could be expanded to provide a more detailed context.

Answer: Done! we improved the introduction also considering other Reviewer’s requests

Reviewer: Methods and Materials: Information regarding the number of patients involved in the register is missing. Details regarding data filtration and mining processes could be included.

Answer: We revised the material & methods section (red text). Criteria on data mining are reported in section 2.1.; We developed description of “surgical” criteria. We also added section 2.2 with better identification of the endpoints.

Reviewer: Section 2.2: It would be beneficial if the authors specified the software and version used for statistical and error analyses.

Answer: Section 2.2 is now 2.3. We added the software specification: SPSS package 18.

Reviewer: Figures 1-3: These figures appear too small for clear viewing. Enlarging them, if feasible, would improve readability.

Figure 2: Consider combining Figure 2(a,b) with Figure 2(c,d) or labeling Figure 2(c,d) as a separate figure. Note that the resolution of Figure 2(c,d) is lower than Figure 2(a,b).

Duplicate Figure 3: The manuscript contains two Figure 3s, which needs clarification or correction.

Answer: We have enlarged the figures and have achieved a better quality. Also, we have revised their numeration.

Reviewer: Conclusion: Mentioning future work related to this study in the conclusion would be beneficial.

Answer: We agree. We developed the paragraph “Limitation of the study” which became “Limitation of the study and future developments”. Here we state our future research which will be published as soon as Meta-gastro will enrol an adequate number of cases to allow the specific evaluations.

Round 2

Reviewer 1 Report

Comments and Suggestions for Authors

Comment

This study is valuable because it presents real-world data from Italy. However, quantitative/anatomic aspect of oligogenic gastric cancer is difficult to understand, making it difficult to understand how this content can be useful to readers. Therefore, this study does not meet the criteria for publication in this journal.

Reviewer 2 Report

Comments and Suggestions for Authors

I accepted the modifications from the authors as per my comments. The quality and presentation of the manuscript are improved.

Comments on the Quality of English Language

No big problem for the English.